# Parameter Selection for a Microvolume Electrochemical *Escherichia coli* Detector for Pairing with a Concentration Device

**DOI:** 10.3390/s19112437

**Published:** 2019-05-28

**Authors:** Evelina J. Y. Han, Kannan Palanisamy, Jamie Hinks, Stefan Wuertz

**Affiliations:** 1Singapore Centre for Environmental Life Sciences Engineering (SCELSE), Nanyang Technological University, 60 Nanyang Drive, Singapore 637551, Singapore; hjyeve@gmail.com (E.J.Y.H.); ktpkannan@mail.zjxu.edu.cn (K.P.); 2School of Civil and Environmental Engineering, Nanyang Technological University, Singapore 639798, Singapore

**Keywords:** *Escherichia coli*, anaerobic respiration, microvolume bioelectrochemical system, detection time, sample concentration

## Abstract

Waterborne infections are responsible for health problems worldwide and their prompt and sensitive detection in recreational and potable water is of great importance. Bacterial identification and enumeration in water samples ensures water is safe for its intended use. Culture-based methods can be time consuming and are usually performed offsite. There is a need to for automated and distributed at-source detectors for water quality monitoring. Herein we demonstrate a microvolume *Escherichia coli* (*E. coli*) detector based on a screen printed electrode (SPE) bioelectroanalytical system and explore to what extent performance can be improved by coupling it with a filtration device. To confidently benchmark detector performance, we applied a statistical assessment method to target optimal detection of a simulated concentrated sample. Our aim was to arrive at a holistic understanding of device performance and to demonstrate system improvements based on these insights. The best achievable detection time for a simulated 1 CFU mL^−1^ sample was 4.3 (±0.6) h assuming no loss of performance in the filtration step. The real filtered samples fell short of this, extending detection time to 16–18 h. The loss in performance is likely to arise from stress imposed by the filtration step which inhibited microbial growth rates.

## 1. Introduction

Improved microbial sensing and monitoring are desirable for a number of environmental applications, including microbial water quality monitoring [1]. In microbial water quality monitoring, the presence and quantity of organisms such as *Escherichia coli* or *Enterococcus faecalis* is indicative of faecal contamination, its probable extent, and risks associated with the water’s intended use. Conceptually, microbial water-quality monitoring falls into two categories based on the different analytical challenges associated with monitoring drinking water [2] or water for recreational use [3]. Microbial quality monitoring of drinking water requires the detection of fewer bacteria (as low as one cell in 100 mL of water) than does recreational water quality monitoring (typically hundreds of cells in 100 mL of water). Determining the safety of drinking water, therefore represents a formidable analytical challenge. Detection of a single cell in 100 mL volume is equivalent to detecting 1 × 10^−14^ g L^−1^ of a mixture of analytes in an undefined environmental matrix [4]. Culture- based techniques are able to detect such low cell numbers, but they do so in time frames (>24 h) that are not compatible with the current advancement towards distributed and remote monitoring in the water industry [5].

Microbial water quality monitoring tests usually take a day. The average residence time of water in a distribution network is less than one day [6]. This creates a conundrum for water companies whose product is usually consumed before tests can definitively determine that it is safe [7]. Water quality monitoring therefore demonstrates a commitment to good practice and a statistical likelihood of safety, but in reality, it does not allow proactive monitoring of microbial water quality. To this end, automated microbial water quality sensors are being developed [7,8]. To overcome the protracted analysis times that result from the stringent water quality metrics, sample concentration has been proposed to reduce analysis time. Microfluidic water concentrator devices able to reduce large volumes of water (reportedly up to 1 m^3^ (Fluigen.com) into small filtrate volumes (≈1–200 µL) quickly (<1.0 h) are appearing on the market. Coupling microvolume detectors with technology able to concentrate water by several orders of magnitude could potentially bring water quality monitoring in-line with current industry needs [7].

Electrochemical methods, including those that use screen printed electrodes (SPEs), are gaining in popularity [9,10,11]. Specific bioelectroanalytical *E. coli* detection utilising defined substrate technology (DST) designed for electrochemical analysis was recently reported [12]. However, in the system previously reported, standard electrochemical cells were used to demonstrate the bioelectroanalytical principle. These systems had a volume of 10 mL and are research tools that are not appropriate for field application. Here we build on the previously reported bioelectrochemical approach and explore its application in a microvolume *E. coli* detector (0.5–1.0 mL) using SPEs. Our aim is to apply a statistical assessment method to evaluate the operating conditions and performance of the microvolume detector, targeting detection of a simulated concentrated water sample [13]. This baseline performance will then used as a comparator for concentrated samples obtained from a commercially available concentration device [14]. We hope to arrive at a broad understanding of device performance limitations and to derive operational insights based on this analysis. To the best of our knowledge, this is the first report of a microvolume bioelectroanalytical SPE *E. coli* detection system. The combination of a statistical assessment tool along with coupling the detector to a commercial concentration device has not been reported elsewhere.

## 2. Materials and Methods


*Growth Conditions and Strains*


Overnight growth of *E. coli* DSMZ 1576 (an environmental *E. coli* isolate [15]) was obtained by inoculating single colonies grown on LB Miller plates into modified M9 minimal media (M9 salts, CaCl_2_, MgSO_4_, thiamine HCl) [9], and incubating for 18 h at 37 °C in a 200 rpm shaking incubator. d-galacturonic acid (3.75 g L^−1^) and glucose (2 g L^−1^) were used as carbon source for modified minimal media M9-2 and M9-G, respectively. Overnight bacterial cultures were centrifuged, the supernatant discarded, and the pellet washed in phosphate buffered saline (PBS) three times before resuspension in modified M9 minimal media and adjusting to the desired inoculum density (CFU mL^−1^). To enable *E. coli* to communicate with the electrode, 2-hydroxy-1,4-naphthoquinone (HNQ) or the peracetylated methyl ester of 2-hydroxy-1-4-napthoquinone-β-D-galactopyranoside (HNQG) were added at a final concentration of 50 µM as previously reported in Hinks et al. [12]. *E. coli* SHSP18 [16] was obtained from The Coli Genetic Stock Center (#4678, CGSC, Yale, CT, USA). M9 minimal media with glucose and 1% (v/v) LB Miller broth (M9-G-LB) was used for overnight culture and experimental studies with the *E. coli* SHSP18. To restore the defective respiratory chain in this mutant, 5-aminolevulinic acid (ALA) was supplemented into the above minimal media with a final concentration of 20 µg mL^−1^ to compare device performance between fermentative and respiratory growth.


*Preparation of Microvolume Electrochemical Reactors*


The caps of 1.5 mL Eppendorf tubes were trimmed off and a 1 mm diameter hole was punctured at the conical portion of the Eppendorf tube using a 30G needle. After sterilisation by autoclave (120 °C, 15 psi, 15 min), the modified tubes were attached with epoxy resin to SPEs (Metrohm Dropsens, Asturias, Spain), which were sterilized by soaking for one minute in 70% ethanol, followed by air drying and ultra violet sterilisation for 20 min. Reactors were assembled in a biological safety hood and the epoxy allowed to cure for a minimum of 12.0 h before use.


*Removal of Oxygen in Mini Reactors*


For the Design of Experiment (DOE) setup, oxygen was removed by sparging the media and headspace with N_2_ gas followed by sealing the only inlet with tape before the start of any electrochemical experiments that prepared and run on bench tops at 37 °C. All reactors were prepared in the anaerobic chamber to purge residual oxygen from all parts of the system. Determination of best detection time and studies with respiratory mutants were conducted entirely in the anaerobic chamber.


*Potentiometric Measurements*


The screen printed electrodes (SPEs) (Metrohm Dropsens) consist of a circular working electrode (4 mm diameter) of either carbon ink or carboxylated carbon nanotubes (CNTs). The SPEs have plain carbon counter electrodes and silver (Ag) as a reference electrode. Potentiometric measurements were performed with VSP-150 multichannel potentiostat (Bio-Logic SAS, Seyssinet-Pariset, France) or DRP-STAT8000 mini potentiostat (Metrohm Dropsens) for experiments run under completely anaerobic conditions. All electrochemical potential values are reported with respect to Ag reference electrode, and average current was recorded every 60 s during chronoamperometric measurements. Chronamperometric analysis was performed immediately upon introduction of a standardised *E. coli* inoculum into a volume of 0.5 or 1.0 mL. Current was collected continuously until a detection event was recorded, defined here when current increase exceeded five times the standard deviation of the baseline current, and example of a raw output can be seen in Figure 1 and exact operating conditions for the 32 runs used to benchmark the system can be found in Table 1.


*Design of Experiments (DoE) Setup and Analysis*


To understand the main effects contributing to detection time, we applied a duplicated DoE half-factorial experimental design generated using Minitab^®^ Statistical Software (Minitab Inc., State College, PA, USA) with five factors set at two levels each, i.e., a 2^(5−1)^ design with repeats randomised in blocks of four according to the design matrix in Table 1. Five easily controllable experimental variables were identified for investigation aimed a striking a balance between a comprehensive but experimentally achievable study. For the settings, we selected two factors that were non-continuous variables: electrode material (plain carbon SPEs and carbon nanotube (CNT) coated SPEs) and the mediator (unconjugated vs. conjugated quinones). The selected electrodes have superficially similar performance but different price points while the conjugated status of the conjugated redox mediator will help address the dynamics of activating the redox moiety. In addition, there were three continuous variables; voltage (200 vs. 600 mV), reactor volume (0.5 vs. 1.0 mL), and inoculum size (500 vs. 5000 CFU). The output determined here was detection time, expressed in h and was determined using as previously reported technique [9], defining a detection event when the sensor registers a current that deviates from several standard deviations of the baseline. Results were analysed by Minitab^®^ multi-factor ANOVA tool.


*Concentration Protocol*


Filter concentrated samples were obtained by passing 5 L of PBS spiked with a known quantity of *E. coli* through an InnovaPrep^®^ system (manufacturer, Drexel, MO, USA). The recovery rate was determined by introducing different fractions of a standardised inoculum of washed overnight *E. coli* culture into PBS which was then eluted through the InnovaPrep^®^ system into an indeterminate volume of less than 1.0 mL. The filtrate was subsequently made up to 1.0 mL volume. The recovery rate of the concentration step was determined by comparing the optical density of the initial spike sample at the time it was prepared with the optical density of the filtrate. A recovery rate of 100% would be when the optical density of the initial sample and the eluent matched. The CFU in the sample spike and the filtrate were subjected to a quality control step by cell counting using the drop plate method [17].

## 3. Results and Discussion

Bioelectroanalytical detection of microbes is analogous to enzyme substrate techniques that utilise enzyme specific glycosides usually conjugated to a chromophore that is activated by contact with a specific enzyme indicating of the presence of a target microbe. In the current study, the detection compound is the peracetylated methyl ester of the naphthoquinone glycoside, HNQG, that yields an electrochemically active HNQ reporter upon contacting galactosidases expressed by *E. coli* [12] and which can be detected at an electrode (Figure 1). The current generated approximates the metabolic activity of the target bacterium and the lag in current pickup is analogous to lag phase and is a function of inoculum size [18,19]. The study is divided into two parts: the first is aimed at applying a statistical technique to benchmark detection times for a novel microvolume electrochemical detector. The second part of the study will assess the impact on detection time when applying a concentration step to spiked samples.


*Statistical Analysis of Operational Factors–Main Effects*


The factors that most influence detection time of the bioelectrochemical system are shown in the main effects plot (Figure 2a). The number of bacteria in the system has the greatest effect on detection time, essentially verifying that this is an *E. coli* detection system (Figure 2a). More specifically, it shows that normalised detection time was reduced from about 7.3 h to around 6.0 h for a one-log increase in inoculum size from 500 to 5000 CFU.

Put into context, concentration of samples contaminated with 1–10 CFU mL^−1^, would require a 2 to 3 log increase in cell number to achieve desirable detection within one working day [19]. To achieve the 6.0 h detection time discerned from the DoE studies would require a sample volume of around 5 L at 1 CFU mL^−1^ and a volumetric concentration factor of 5000× and 10000× to achieve detection volumes of 1.0 mL and 0.5 mL, respectively. Such large concentration factors are manageable with the proposed filtration technique which can concentrate 5 L to less than 1 mL in around 0.5 h. Voltage had the second largest influence on detection time (Figure 2a). By setting the electrode potential at 600 mV, the detection time increased by 50 min compared to operating at 200 mV (Figure 2a). An applied potential of 200 mV is commonly agreed to be compatible with biological electron transport processes [20]. The rationale for setting the detector at 600 mV was to provide a greater electrochemical gradient for HNQ oxidation. On the contrary, the exact opposite appears to be true and the oxidising conditions imposed by a higher poise negatively affected this detection system.

The remaining three factors tested influenced detection time in the order: volume > mediator > electrodes (Figure 2a). All three were below the Pareto cut off value, in this sense they can be deemed as factors that contribute to the noise of the technique, which we wish to minimise in order to achieve better reproducibility (Figure 2b). Essentially, more than 80% of the performance is determined by operating the detector at 200 mV and with an inoculum size of 5000 cells.

The influence on detection time for two of the factors that were below the Pareto threshold (electrode type and reactor volume) were not intuitive in both magnitude and direction of effect. For example, the influence of the electrode type was determined to be relatively minor. Electrode modification is commonly applied to achieve performance improvements in bioelectroanalytical systems [21] and such a finding forces conceptual reassessment of the bioelectroanalytical *E. coli* detector. A detection event occurs at early log phase, when the cell number is still low, and before an electroactive biofilm has had time to form. In this sense, the analyte is not *E. coli* but reduced HNQ which is cleaved from the glycoside detection compound by *E. coli* β-galactosidase, so in the strictest sense, the system here is as much an electrochemical detector as it is a bioelectrochemical one; a hybrid so to speak which may have implications for further optimisation strategies.

Similarly, larger reactor volumes gave rise to shorter detection times which is inconsistent with the idea that these unstirred systems would undergo significant mass transfer limitations and that smaller volume detection chambers would realise better detection times [22]. The power of the statistical technique allows further examination of how factors influenced system performance by revealing interactions that adversely contribute to the error of the technique.


*Statistical Analysis of Operational Factors–Interactions*


Aside from looking at the performance effects of each factor singly, interesting insights into system behaviour can be gleaned from looking at the influence of the interactions between two factors and how this influences system performance.

An interaction exists between the mediator type and the inoculum size. The interaction plots in Figure 3 can be interpreted by the extent to which lines converge or diverge. Lines running parallel to one another indicate the absence of an interaction between two factors; diverging, converging or crossing lines indicate interactions with the slope of the lines indicating the strength and direction of this effect. When an interaction exist between two factors, this technique reveals the settings which will realise performance improvements when the two factors are applied to a given system. The conjugated mediator, HNQG, has a stronger positive influence on detection time compared to the unconjugated mediator, HNQ, when the inoculum size is high (5000 CFU, Figure 3a). The reason for this is not clear but the nature of the interaction allows us to rule out the inhibition of *E. coli* by HNQ, something that has been reported before [23] as the line for the effect of inoculum size, remains flat at 500 CFU demonstrating the near absence of a mediator effect at low inoculum size (Figure 3a). In fact the interaction is influenced entirely by HQNG with the higher inoculum size. This interaction, and the effect of the mediator in isolation, is positive as HNQG is essential to selectively identifying *E. coli* in the sample, the fact that it does so along with performance gains is desirable.

Whilst the electrode material in isolation had unremarkable effect on performance, it was implicated in a number of interactions in particular with inoculum size and with voltage. The electrode material had an appreciable impact on detection times when the voltage was low and when the inoculum size was high (Figure 3b,c). This effect was driven by the CNT electrodes as the detection time remained similar for both conditions when the effect of the plain carbon electrode was considered. Such interactions highlight the power of the statistical technique and its ability to reveal system behaviour that would not be uncovered by conventional investigations. In this instance, dismissing the effect of electrode type on performance is a risk when considering this factor independently of others and, while in this instance, electrode type would not realise large performance gains, the CNT electrode is likely to minimise noise in the system.

The final interaction worthy of mention is the interaction between electrodes and volume (Figure 3d), which combined with the effect of the volume in isolation, is noteworthy as the direction of effect is different to which was expected for reasons noted above. Suspecting oxygen diffusion into the reactor has a more pronounced affect in the smaller volume reactor, we used *E. coli* strains with incomplete respiratory chains to determine if respiration or fermentation was driving the bioelectrochemical reaction in the detector. Armed with such information, it may be possible to improve performance by tweaks to the formulation of the medium - for example by increasing the quantity of fermentable substrates or by the addition of an oxygen scavenging reductant. In this sense the statistical study, in particular the analysis of interactions, provided clues to uncontrolled factors that affect system performance and offering opportunities to make improvements.


*Performance Considerations—Best Achievable Detection Time*


The statistical study revealed the best operating conditions for the sensor ordered by magnitude of effect: high inoculum density > 200 mV poise > 1.0 mL volume > HNQG mediator > CNT electrodes. Using these conditions, we were interested to examine to what extent the metabolic strategy employed by *E. coli* (fermentation vs. respiration) impacts its interaction with the electrode. *E. coli* SHSP18 has a defective respiratory chain that can be restored by supplementation with ALA [24,25]. SHSP18 was only able to initiate a detection event for 5000 CFU at 13.8 h in the absence of ALA compared to 6.6 h when the defective respiratory chain was restored by supplementing the medium with 20 µg mL^−1^ ALA (Figure 4). This implicates the dominance of respiratory behavior in reducing the electrode over fermentation and suggests that further improvements can be realised by limiting the presence of competing electron acceptors, such as oxygen, inthe detector. We therefore used the anaerobic chamber and determined the best possible performance we could achieve with the current set up was 4.3 h (± 0.6) (Figure 4). This would equate to an ideal detection time of around 5.0 h (including filtration) for a sample of containing 1 CFU mL^−1^ assuming it is possible to achieve sample concentration without a loss of performance. This would be an improvement on standard techniques which require 24–48 h for analysis and fits well with the ideal of completing an analysis within one working day [19,26,27].


*Performance—Sample Concentration*


The InnovaPrep^®^ concentrating pipette was simple to operate and was able to achieve a concentration factor of c.a. 5000× from a spiked 5 L sample in <0.5 h using both 0.20 and 0.45 µm filters. Table 2 shows that that recovery was excellent (c.a. 100%) when the spike was not fractionated. However, reducing the spike concentration adversely affected the relative recovery of *E. coli*. This phenomenon has been previously observed for filtration-based microbial concentration techniques [28].

Despite good recovery of spiked samples in the eluent, the detection times for filter concentrated samples were poor. Figure 5 shows that the detection time for 5000 CFU was c.a. 16.0 and 14.0 h for those samples recovered using 0.20 μm and 0.45 μm filter, respectively. Compared to a best possible detection time of 4.3 h (±0.6), and an equivalent detection time of 6.3 h (±0.1) (Figure 4) that can be achieved without filtration and using HNQ. It is evident that the filtration step has a significant and adverse effect on detection time and whilst still in line with current techniques, they do not represent an improvement that justifies the increased complexity of the approach.

A simple explanation for the observed adverse effect of the eluent would be that the filtration step reduced the viability of the recovered sample. However, prior to submitting the recovered samples to the detector, the inoculum size was confirmed by plate counts showing excellent viability and recovery on solid medium. This points to a difference in viability between cells grown on agar and those in a liquid medium suggesting that the filtration step renders the cells less able to tolerate osmotic stress or the specific conditions of the detector, which on account of the electrode or the mediator may apply an unconventional stress to the test organism. Shear stress implied during filtration is well documented and there are various techniques for recovery of injured microbes [29]. Even though liquid recovery of injured bacteria has been described [30], many methods exist to recover injured cells whichvary depending on the nature of the injury [31]. However, liquid recovery steps prior to agar plating usually do not use conventional medium but instead a recovery medium with various additives such as catalase or pyruvate, and it is well know that selective media adversely effect the recovery of damaged cells [31]. The medium used for downstream bioelectroanalytical detection can be deemed a selective medium on account of the enzyme specific detection compounds it contains and its minimal medium base.

The InnovaPrep^®^ system uses a Wet Foam Elution™ backwash to recover cells from the filter. This contains phosphate buffered saline (PBS) and 0.075% Tween 20. As the detection system relies on growth rate to achieve a critical current threshold, it is conceivable that the surfactant contributes to the diminished performance observed in the filter recovered samples. Growth analysis of *E. coli* after being subjected to a washing step and treatment with 0.075% Tween in PBS for 45 min showed around a 50% reduction in growth rate (0.27 h^−1^) compared to a PBS control (0.43 h^−1^) (Figure 5b). Whilst it is unlikely that there will be no performance pay off for filtration, the performance cost observed here is untenable and is, at least in part, owing to the effect of Tween 20 in the eluent - which is known to perturb membranes - or the washing step itself [32]. All of the Tween family of non-ionic surfactants have been shown to inhibit the growth of *E. coli* with Tween 20 exerting the most significant effect with it I also implicated in preventing biofilm formation of *E. coli* [33,34]. Whilst beyond the scope of the current study, it is likely that reformulation of the eluent tailored to the specific needs of the detection system could lead to significant performance benefits. Interestingly Tween 80 has been shown to aid in the recovery of damaged cells and even in enhancing microbial charge transfer in bioelectrochemical systems [31,35], thus the formulation of the medium in conjunction with the eluent needs addressing further. Finally, our technique relies on lag phase for enumeration whereas plate counts allow direct assessment of viability. Plate counts are less sensitive to cell damaged than our technique which may partly explain the apparent differences in post concentration viability between solid and liquid media. 

## 4. Conclusions

The statistical technique applied here is useful in aiding parameter selection for a novel detection device in order to benchmark performance. Such insights enable subsequent impact assessment of operational interventions, in this instance a concentration step. The statistical technique allowed confident parameter selection, and revealed additional performance insights like how competition between the electrode and other terminal electron acceptors and also detector volume impact performance. Respiratory behavior accounts for the major component of system performance rather than fermentation and conditions must be kept strictly anaerobic to improve detection. Anaerobic conditions can likely be achieved with the addition of a reductant like cysteine hydrochloride. Even though a robust estimate of detection time was determined and excellent concentration times were demonstrated, the impact of the concentration step led to protracted detection times. The detection times achieved after concentration (≈16 h) were close to that of standard techniques (24 h) rendering the coupling of the two techniques incompatible at this juncture due to increased costs and complexity. Poor detection times following concentration likely results from Tween 20 in the eluent. The rapid concentration time and large concentration factors realised by the InnovaPrep^®^ make it an operationally attractive proposition to concentrate water samples with low numbers of organisms. However, the eluent negatively affects downstream detection times and its reformulation may add value to the proposed detection framework. The filtration concept should be revisited with the eluent reformulation as a priority research and development opportunity. While the goal of rapid drinking-water quality monitoring remains out of reach for the proposed set up, the microvolume detectors would be suitable for environmental water-quality monitoring.

## Figures and Tables

**Figure 1 sensors-19-02437-f001:**
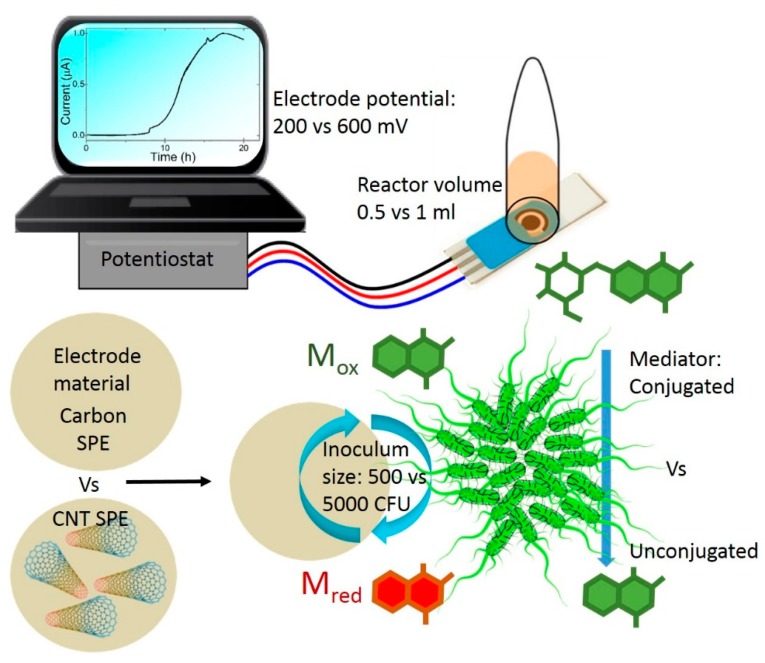
Conceptual experimental approach applied in the Design of Experiments (DoE) study which examined the effect of electrode potential, reactor volume, electrode material, inoculum size, mediator type on detection time. DoE experimental designs typically involve holding each factor at a high or low level. For non-continuous variables, in this case the electrode material and mediator type, ‘high’ and ‘low’ were arbitrarily assigned. A representative raw data output for each of the 32 DoE trials is shown in the laptop graphic on the top left.

**Figure 2 sensors-19-02437-f002:**
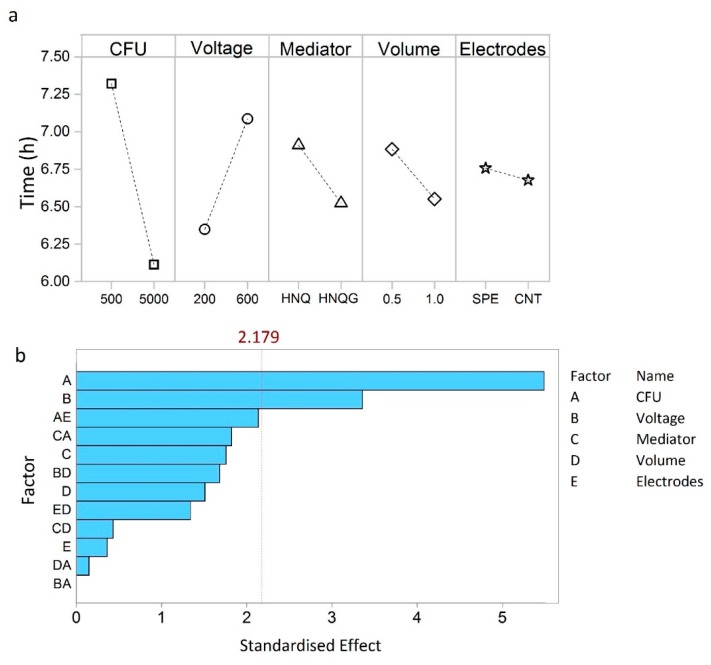
Main effects plot (**a**) showing which factors have the greatest effects on detection time. Pareto chart (**b**) showing the most significant factors and interactions. The dotted line indicates the standardized effect cut off value (2.179), factors having an effect below this do not contribute significantly to detection time.

**Figure 3 sensors-19-02437-f003:**
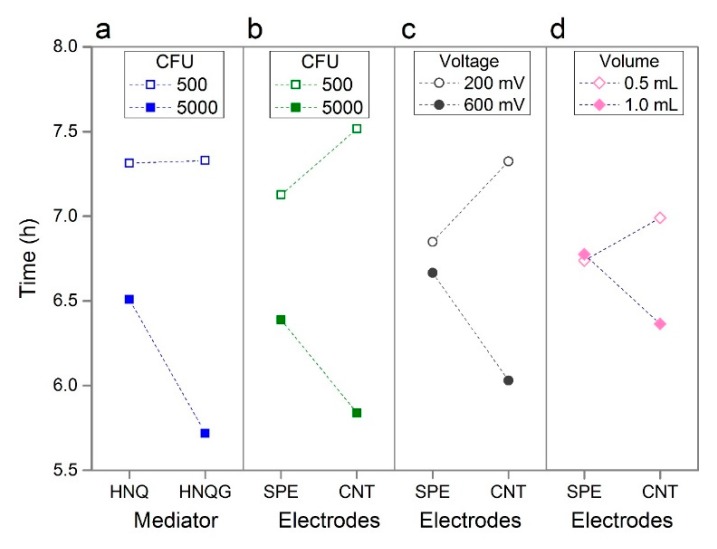
Interaction plots showing four prominent interactions: Mediator vs. Inoculum Size (**a**); Electrodes vs. Inoculum Size (**b**); Electrodes vs. Voltage (**c**); Electrodes vs. Volume (**d**).

**Figure 4 sensors-19-02437-f004:**
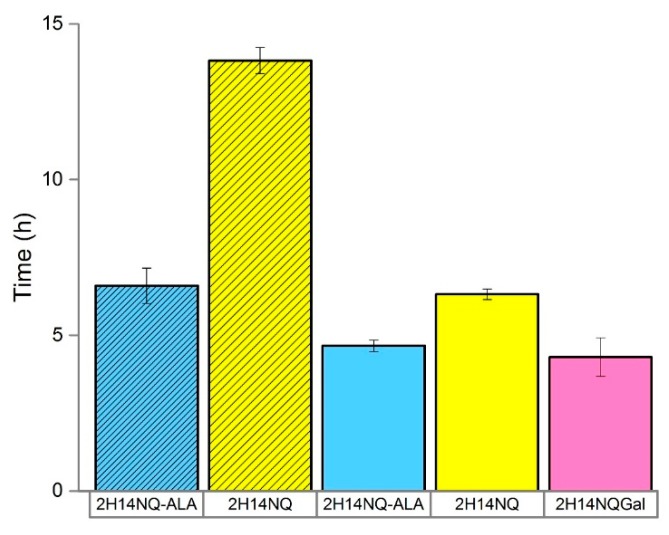
**Detec**tion time (h) for two *E. coli* strains; SHSP18 (shaded) and DSMZ-1576 (unshaded) in the presence and absence of ALA. Error bars represent the range of detection time between replicates (n = 2–3).

**Figure 5 sensors-19-02437-f005:**
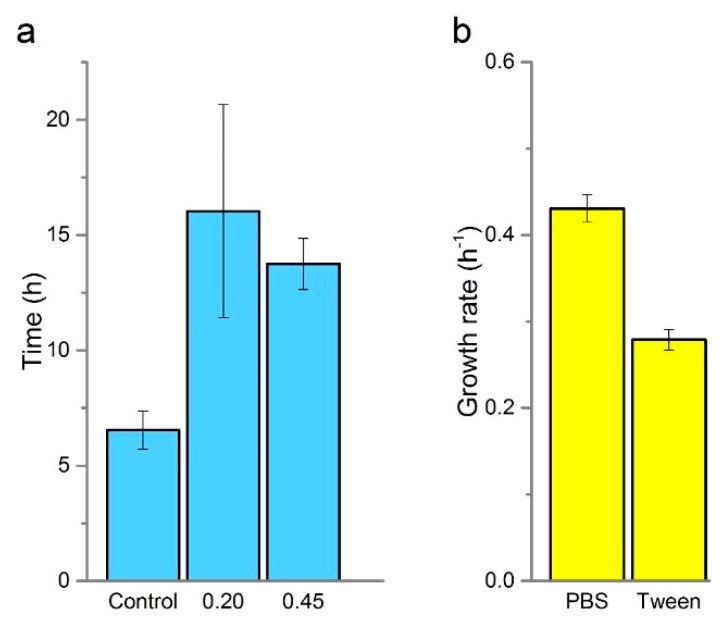
Detection time using the eluent obtained from the InnovaPrep^®^ concentrator, a 5 L sample **c**ontaining 5000 CFU of *E. coli* DSMZ 1576 was reduced to a volume of less than 1.0 mL and subsequently made up to volume (1.0 mL) (**a**). The effect on growth rate of *E. coli* DSMZ 1576 after being treated with Tween 20 for 45 min (**b**). Error bars represent the detection time range between replicates (n = 2–3).

**Table 1 sensors-19-02437-t001:** Experimental design matrix and raw output data (detection time) generated using the DOE feature in Minitab^®^. The following five factors were set either low or high consecutively: Inoculum, 500 and 5000 CFU; voltage, 200 and 600 mV; mediator, HNQ and HNQG; volume, 0.5 and 1.0 mL; electrodes, SPE and CNT. The experiments were ran in random blocks of four.

Run Order	Standard Order	Factors					Detection Time (h)
		CFU	Voltage	Mediator	Volume	Electrodes	
1	32	High	High	High	High	High	6.09
2	30	Low	High	High	Low	High	8.39
3	31	Low	Low	Low	High	Low	7.80
4	29	High	Low	Low	Low	Low	6.67
5	25	Low	Low	High	Low	Low	7.07
6	27	High	Low	High	High	Low	5.92
7	28	Low	High	Low	High	High	7.54
8	26	High	High	Low	Low	High	6.99
9	16	High	High	High	High	High	6.3
10	15	Low	Low	Low	High	Low	6.05
11	13	High	Low	Low	Low	Low	7.52
12	14	Low	High	High	Low	High	8.49
13	7	Low	Low	High	High	High	6.65
14	5	High	Low	High	Low	High	5.84
15	6	Low	High	Low	Low	Low	6.9
16	8	High	High	Low	High	Low	8.02
17	19	High	Low	Low	High	High	4.89
18	20	Low	High	High	High	Low	7.17
19	18	High	High	High	Low	Low	5.12
20	17	Low	Low	Low	Low	High	7.57
21	21	High	Low	High	Low	High	4.75
22	23	Low	Low	High	High	High	6.29
23	24	High	High	Low	High	Low	6.12
24	22	Low	High	Low	Low	Low	7.44
25	3	High	Low	Low	High	High	5.10
26	1	Low	Low	Low	Low	High	7.15
27	4	Low	High	High	High	Low	7.55
28	2	High	High	High	Low	Low	6.47
29	12	Low	High	Low	High	High	8.05
30	9	Low	Low	High	Low	Low	7.02
31	11	High	Low	High	High	Low	5.27
32	10	High	High	Low	Low	High	6.75

**Table 2 sensors-19-02437-t002:** Recovery efficiency of E.coli DSMZ 1576 from a spiked 5 L sample using 0.20 and 0.45 μm pore size filters passed through the InnovaPrep© concentrating pipette. The recovery efficiency was determined by the extent to which the optical density of the spike and that of the eluent matched.

Filter Pore Size (μm)	Spike Volume (mL)	% (CFU mL^−1^ Recovered)
Control	NA	133%
0.20	1	107%
	0.5	73%
	0.25	106%
	0.125	50%
0.45	1	108%
	0.5	60%
	0.25	65%
	0.125	74%

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
