# Peer review of "Parameter Selection for a Microvolume Electrochemical Escherichia coli Detector for Pairing with a Concentration Device"

_sensors, 2019, doi:10.3390/s19112437_

Round 1
Reviewer 1 Report
General Comments:
This paper is about parameter selection for a microvolume electrochemical Escherichia coli detector.
The authors try to solve water treatment management challenges in context of bacterial identification and also simplyfy work with enumeration in water samples ensures water, that safe for its intended use. The paper discusses the microvolume Escherichia coli (E. coli) detector based on a screen-printed electrode (SPE) bioelectrochemical system and explore to what extent performance can be improved by coupling it with a filtration device to improve performance. The paper is interesting, well-written and full of information. However, a few minor changes, below, needed to be done before publishing the paper.
Minor Comments:
a. The reference list is too short and contains only a few recent literature position and must be improved in order to provide sufficient background and include all relevant references
b. At various places some sentences could be found that are not good and moderate English changes required. Please read all text again and try to repeat editing of English language and style again and again.
Specific comments:
a. The
introdution is too short and contains only a few recent literature
position and lack of information that should be included about
development of proposed technology for a microvolume electrochemical Escherichia coli detector. Please improved
b. I believe that the Author should make an effort in carefully presenting the material/methods and achived results for in better explain how the proposed technology for a microvolume electrochemical Escherichia coli detector was performed, how it was implemented and what is the future perspective of this system could look like in comparation with others what has been developed till now and try to enhance discussion in whole manustript, avoiding verbosity and carefully checking the grammar. I found the manuscript confusing and probably because of that also the results seem questionable
c. The Conclusion is too general please specify and improve this part, and try to avoid "we" have demonstrate... it was demonstrate etc.
d. Page 11, line 396 Change “number” of references by add “28” now is wrong and must be improved
Reviewer 2 Report
1. In the manuscript, the authors indicated that the best achievable detection time for a simulated 1 CFU ml-1 sample would be 4.3 (± 0.6) h, this result was assumed that no loss of performance in the filtration step. However, in reality, if the loss in performance from stress that imposed by the filtration step was taken into account the actual detection time would be 16-18h. This result is no big different to standard techniques which require 24 h for analysis.
2. The authors indicated that with current set up and also using of anaerobic chamber to improve performance of detection time to be 4.3 h (± 0.6). The authors need to address what were the conditions of current set up in detail and how to apply anaerobic chamber to an on line detector is also suggested.
3. The authors need to show more evidence to prove that the results and discussions presented in this manuscript meet the research goals set for this study.
4. From the conclusions, the author also did not clearly explain the matters needing attention to meet the needs in the study.
